# Attitudes toward Receiving COVID-19 Booster Dose in the Middle East and North Africa (MENA) Region: A Cross-Sectional Study of 3041 Fully Vaccinated Participants

**DOI:** 10.3390/vaccines10081270

**Published:** 2022-08-06

**Authors:** Mohamed Abouzid, Alhassan Ali Ahmed, Dina M. El-Sherif, Wadi B. Alonazi, Ahmed Ismail Eatmann, Mohammed M. Alshehri, Raghad N. Saleh, Mareb H. Ahmed, Ibrahim Adel Aziz, Asmaa E. Abdelslam, Asmaa Abu-Bakr Omran, Abdallah A. Omar, Mohamed A. Ghorab, Sheikh Mohammed Shariful Islam

**Affiliations:** 1Department of Physical Pharmacy and Pharmacokinetics, Poznan University of Medical Sciences, Rokietnicka 3, 60-806 Poznan, Poland; 2Doctoral School, Poznan University of Medical Sciences, 61-701 Poznan, Poland; 3Department of Bioinformatics and Computational Biology, Poznan University of Medical Sciences, 61-701 Poznan, Poland; 4National Institute of Oceanography and Fisheries (NIOF), Cairo 11694, Egypt; 5Health Administration Department, College of Business Administration, King Saud University, P.O. Box 71115, Riyadh 11587, Saudi Arabia; 6Department of Cell Biophysics, Faculty of Biochemistry, Biophysics and Biotechnology, Jagiellonian University in Kraków, 31-007 Krakow, Poland; 7Physical Therapy Department, Jazan University, P.O. Box 114, Jazan 45142, Saudi Arabia; 8Oral Health and Promotion Unit, Faculty of Dentistry, Al-Quds University, Jerusalem P.O. Box 89, Palestine; 9Dental College, AlNoor University College, Bartella 46476, Iraq; 10Faculty of Medicine, Al Neelain University, Khartoum P.O. Box 12702, Sudan; 11Faculty of Medicine for Girls, Al-Azhar University, Cairo 71524, Egypt; 12Department of Pharmaceutical Services and Sciences, Children’s Cancer Hospital Egypt (CCHE-57357), Cairo 11617, Egypt; 13Wildlife Toxicology Laboratory, Department of Animal Science, Institute for Integrative Toxicology (IIT), Michigan State University, East Lansing, MI 48824, USA; 14Institute for Physical Activity and Nutrition (IPAN), School of Exercise and Nutrition Sciences, Deakin University, Melbourne, VIC 3125, Australia

**Keywords:** coronavirus, vaccine hesitancy, SARS-CoV-2, vector vaccines, mRNA vaccines

## Abstract

COVID-19 vaccines are crucial to control the pandemic and avoid COVID-19 severe infections. The rapid evolution of COVID-19 variants such as B.1.1.529 is alarming, especially with the gradual decrease in serum antibody levels in vaccinated individuals. Middle Eastern countries were less likely to accept the initial doses of vaccines. This study was directed to determine COVID-19 vaccine booster acceptance and its associated factors in the general population in the MENA region to attain public herd immunity. We conducted an online survey in five countries (Egypt, Iraq, Palestine, Saudi Arabia, and Sudan) in November and December 2021. The questionnaire included self-reported information about the vaccine type, side effects, fear level, and several demographic factors. Kruskal–Wallis ANOVA was used to associate the fear level with the type of COVID-19 vaccine. Logistic regression was performed to confirm the results and reported as odds ratios (ORs) and 95% confidence intervals. The final analysis included 3041 fully vaccinated participants. Overall, 60.2% of the respondents reported willingness to receive the COVID-19 booster dose, while 20.4% were hesitant. Safety uncertainties and opinions that the booster dose is not necessary were the primary reasons for refusing the booster dose. The willingness to receive the booster dose was in a triangular relationship with the side effects of first and second doses and the fear (*p* < 0.0001). Females, individuals with normal body mass index, history of COVID-19 infection, and influenza-unvaccinated individuals were significantly associated with declining the booster dose. Higher fear levels were observed in females, rural citizens, and chronic and immunosuppressed patients. Our results suggest that vaccine hesitancy and fear in several highlighted groups continue to be challenges for healthcare providers, necessitating public health intervention, prioritizing the need for targeted awareness campaigns, and facilitating the spread of evidence-based scientific communication.

## 1. Introduction

The coronavirus disease 2019 (COVID-19) outbreak has prompted remarkable research responses in various fields [1,2,3,4,5]. The massive financial and organizational support allowed scientists and researchers to develop many vaccines with different technologies. Most of these vaccines passed through preclinical and clinical trials and have been approved at an extraordinary speed [6]. This pandemic reminds humanity of the influenza pandemics in 1919, 1958, 1968 and 2009. The world lost more than 40 million people, but the vaccines accompanied by antiviral drugs influenced the severity of the influenza virus and decreased the mortality rate [7]. The first COVID-19 incidents were identified in early December 2019 in China, and the molecular biology of the causative agent, SARS-CoV-2, was defined in January 2020. In December 2020, the earliest vaccine batches were ready to be distributed. More than 3 billion persons worldwide has been given at least the first dose of the COVID-19 vaccine by September 2021 [8].

Six COVID-19 vaccines were approved in the Middle East and North Africa (MENA) region: mRNA-1273 (Moderna, Cambridge, MA, USA), given as two doses with at least 28 days between doses [9]; mRNA vaccine BNT162b2 (BioNTech/Pfizer, Mainz, Germany/New York, NY, USA), given as two doses with at least 21 days between doses [9]; adenoviral vector vaccine AZD1222 (Oxford/AstraZeneca, Oxford, UK/Sodertolje, Sweden), given as two doses with almost 1–3 months between doses, similarly to Gam-COVID-Vac (Sputnik V, Russia) [10]; Ad26.COV2.S (Janssen/Johnson & Johnson, Leiden, The Netherlands/New Brunswick, NJ, USA), given as a single shot [9]; and BBIBP-CorV (Sinopharm, Beijing, China), administered on a 0/21-28-day schedule [11].

The clinical studies supported by legitimate reports and observations showed great effectiveness toward SARS-CoV-2 infection [12,13]. However, the effectiveness declines over time because of two main factors: (i) a drop in IgG antibody serum titers within weeks from the last administered dose [14]; and (ii) the evolution of new SARS-CoV-2 variants that have more vaccine resistance attributes, such as B.1.617.2 (delta variant) and B.1.1.529 (omicron variant). Both variants show higher morbidity and could lead to more severe symptoms, causing higher mortality [15].

During the follow-up process of 6 months, participants in the BNT162b2 vaccine clinical study revealed that effectiveness towards infection declined by around 6% every 8 weeks [16]. However, these findings should be generalized with caution. The investigation could not evaluate the efficacy against the delta variant, which has been prevalent internationally since June 2021. Unfortunately, World Health Organization (WHO) data revealed that different COVID-19 vaccines are less effective for new variants [17]. While vaccination effectiveness toward infection has dropped, boosting could be appropriate for some individuals [18].

The Arab world’s vaccine acceptance level is lower than that worldwide [19]. A study conducted on 870 participants from 22 Arabian countries found that 62.4% of them accepted the COVID-19 vaccine. Another study conducted on 3936 participants from Saudi Arabia, Kuwait, and Jordan found that 38.9% of participants would accept the COVID-19 vaccine, which indicates a serious problem in fighting the COVID-19 pandemic [20]. However, the concerns of the public may increase during the current omicron surge, and their attitudes could change [21].

According to Bartsch et al., achieving public herd immunity requires vaccinating 70% of the population, and that in turn relies on the willingness of the public to accept it [22].

Thus, assessing willingness to receive a COVID-19 booster dose is vital. To the best of our knowledge, no such reports or studies have been conducted in the MENA region. Hence, this study aimed to assess the attitudes of fully vaccinated adults in the MENA region toward receiving a COVID-19 booster dose and identify the related hesitation variables that contributed to their decisions.

## 2. Materials and Methods

### 2.1. Procedure

An online survey was conducted in the MENA region in November and December 2021. After translating from English to Arabic, we used a validated survey (Cronbach’s alpha showed good reliability, α = 0.82–0.93) originally designed by Rzymski et al. [17] after obtaining consent. The survey distribution started on 6 December 2021 and was concluded on 9 January 2022. The survey was distributed to various groups, including educational, commercial, and community organizations, through social media channels, mainly WhatsApp and Facebook.

The main survey questions included: (Ⅰ) Are you fully vaccinated against COVID-19? (Ⅱ) Which COVID-19 did you receive? (Ⅲ) Please rate the severity of the side effects after receiving the first dose of the COVID-19 vaccine. (Ⅳ) Please rate the severity of the side effects that occurred after receiving the second dose of the COVID-19 vaccine. (Ⅴ) Are you willing to receive a booster dose of the COVID-19 vaccine if it is available? (Ⅵ) Please assess the level of fear associated with receiving the possible additional dose of the COVID-19 vaccine. (Ⅶ) Have you been infected with COVID-19? Moreover, fear and side effects were reported on a 10-point Likert scale (1 is the lowest and 10 is the highest).

### 2.2. Study Population

Participants had to be Arabic speakers, have received the first and second dose of COVID-19 vaccination (or only one dose in case of Ad26.COV2.S), be living in any MENA region country (Egypt, Jordan, United Arab Emirates, Kuwait, Bahrain, Saudi Arabia, Oman, Qatar, Yemen, Syria, Palestine, Algeria, Morocco, Libya, Tunisia, Iraq, and Sudan), and be at least 18 years old. The online survey was available in Arabic and was designed and hosted on Microsoft Forms. Participation was voluntary, and before filling out the survey, each participant had to confirm their consent to participate. We distributed the survey using social media and mailing lists.

### 2.3. Statistical Analysis

Data were analyzed using TIBCO Statistica (version 13; equipped with Medical Bundle version 4.0.67) and PQStat Software (version 1.8.2). OpenEpi was used to calculate the sample size according to the proportion. The following parameters were used: the population size was 100,000,000, and 5% was the confidence limit. The hypothesized frequency was set to 50%, and the design effect was 1. Therefore, for a 95% confidence level, the estimated minimum sample size was 384 per country. Missing data were treated by listwise and pairwise deletions. Categorical data were reported as frequency/percentage, and continuous data as median (interquartile range, IQR). Normality was calculated using Shapiro–Wilk tests. Differences between side effects and fear among those willing to receive the booster dose were calculated by Mann–Whitney *U* test. Chi-squared test with Bonferroni correction was used to identify significant predictors for fear and willingness to receive the booster dose. Kruskal–Wallis ANOVA was used to associate the fear level with the type of COVID-19 vaccine. Moreover, univariate and multifactorial logistic regression were performed to confirm the results and reported as odds ratios (ORs) and 95% confidence intervals (95% CI). *p*-value < 0.05 was considered statistically significant for all the results.

## 3. Results

### 3.1. Demographic Characteristics

A total of 4056 responses were collected, and only 3041 (75.0%) met the study criteria for further assessments (see Figure 1 for inclusion/exclusion process). The current study’s participants resided in five countries (Egypt, Iraq, Palestine, Saudi Arabia, and Sudan), and their demographic characteristics are summarized in Table 1.

A majority of the participants (i) were below 50 years old; (ii) had higher BMI (51.1%, *n* = 1520); (iii) were settled in urban areas; (iv) had received a bachelor’s degree (i.e., graduate studies); (v) were not infected by SARS-CoV-2; and (vi) had never been administered the influenza vaccine. Approximately 11.9% of respondents (*n* = 361) had at least one chronic disease, with asthma the most common (5.1%, *n* = 154), and there were (9.6%, *n* = 292) immunosuppressed patients. The majority of participants were vaccinated with BNT162b2 (35.2%, *n* = 1071), followed by AZD1222 (30.5%, *n* = 928) and BBIBP-CorV (11.0%, *n* = 335). The frequency of the administered vaccines is shown in Figure 2a.

### 3.2. Willingness to Accept COVID-19 Vaccine Booster Dose

Overall, 60.2% (*n* = 1831) of respondents were willing to receive the COVID-19 vaccine booster dose, while 20.4% (*n* = 619) were hesitant. The principal reasons to refuse the vaccine were safety uncertainties (14.6%, *n* = 443), belief in the non-necessity of the booster dose (14.3%, *n* = 436), and side effects associated with previous COVID-19 vaccine doses (10.3%, *n* = 314).

Individuals earlier vaccinated with AZD1222 were more willing to accept the booster dose (82.0%, *n* = 133), followed by those vaccinated with Ad26.COV2.S (79.1%, *n* = 144) and BBIBP-CorV (77.8%, *n* = 200), while among those vaccinated with BNT162b2, 29.5% (*n* = 267) stated no desire to be vaccinated again.

The willingness to accept a COVID-19 vaccine booster dose was significantly higher in males (*p* < 0.0001), individuals with obesity (*p* < 0.02), those who regularly or irregularly receive influenza vaccination (*p* < 0.001), and who have not been infected by SARS-CoV-2 (*p* < 0.0001). Saudi citizens were more likely not willing to receive the booster dose compared with citizens in other countries in the region (Table 2). No significant differences were observed to influence the willingness to receive the booster dose based on area of residence, status as an immunosuppressed or chronic patient, and level of education.

More serious side effects after the first and second dose were significantly higher among participants unwilling to receive the booster dose than those willing to receive it (Figure 3)—including those who received one dose of Ad26.COV2.S (5 (3–8), *n* = 37; vs. 3 (1.5-6), *n* = 144; respectively; *p* = 0.01, *z* = 2.59).

### 3.3. Fear of COVID-19 Vaccine Booster Dose

On a 10-point Likert scale (Figure 2b), respondents willing to receive the booster dose reported low fear levels with a median of 3 (1.0–5.0), including only 14.6% (*n* = 263) with a fear level above 5. No significant differences were found between the type of COVID-19 vaccine and the fear of booster dose (*p* = 0.15). Participants aged <50 years, females, rural residents, immunosuppressed patients, and Saudi citizens reported higher fear levels (Table 3). Higher fear levels were reported by participants unwilling to receive the booster dose compared to those willing to receive it (Figure 3).

### 3.4. Preferences for Type of Booster COVID-19 Vaccine Dose

The participants willing to receive the COVID-19 booster did not prefer the same type of vaccine as previously. The individual responses regarding a specific COVID-19 vaccination they preferred as a future booster dose are shown in Figure 4a (excluded from the graph: AZD1222-mRNA-1273, *n* = 2; BNT162b2-mRNA-1273, *n* = 2). In general, 9.5% (*n* = 218) said they had no specific interest in the type of vaccine, while 17.8% (n = 410) said they could not decide at the time of the study. Nevertheless, the majority of individuals that fulfilled their first BNT162b2 and Ad26.COV2.S vaccination regimens wanted to get a booster dose with the same type of vaccine, if possible (71.9%, *n* = 564 and 56.6%, *n* = 101, respectively). In the case of mRNA-1273, 54.1% (*n* = 47) of the individuals questioned expressed interest in getting it as a possible booster dose. In contrast, the survey question about the AZD1222 type showed that only 46.6% (*n* = 342) of those questioned expressed interest in receiving it as a booster dose. Overall frequencies of top-selected booster doses appear in Figure 4b.

### 3.5. Predictors of Acceptance of COVID-19 Vaccine Booster Dose

Several factors increase the willingness to accept a booster dose, such as being fully vaccinated with Ad26.COV2.S, AZD1222 or BBIBP-CorV. Those who received the mRNA-1273 vaccine showed less likelihood of accepting the booster dose. Regardless of the received COVID-19 vaccine, the side effects of previous doses and fear were associated with declining the booster dose. Participants infected with SARS-CoV-2 after the first dose were not likely to accept the booster dose. Males, Sudanese, Egyptians, or those with an influenza vaccination history were willing to accept the booster dose. The binary and multifactorial backward stepwise logistics of the significant predictors of accepting a COVID-19 vaccine booster dose are shown in Table 4 and Table 5, respectively.

## 4. Discussion

The majority of the participants (60.2%) were willing to receive a COVID-19 booster dose, according to our findings. Fortunately, it seems that vaccine hesitancy is declining as more information on the vaccine’s safety and effectiveness becomes available. However, there are still certain factors affecting individuals’ decisions whether to receive the COVID-19 booster dose, including fear, side effects, several demographic factors, and previous vaccination with influenza vaccine.

It was reported that the COVID-19 vaccine acceptance rate in Middle Eastern countries was relatively low. Abu-Farha et al. reported that only 24.9% (*n* = 2925) of the population of some Middle Eastern countries agreed to be vaccinated against COVID-19, while 32.6% were hesitant and 42.5% objected [23]. Qunaibi et al., in a study involving 36,220 participants, declared a significant rate of vaccine hesitancy among Arabs in and outside the Arab region (83% and 81%, respectively) [24]. In addition, the rate of compliance with COVID-19 vaccination was lower in Arab countries than the global rate [19,24]. For instance, China scored an acceptance rate of 90% [25], and Canada scored 76.5% [26], while the United States of America was at 69% [27,28,29] and Russia scored 55% [25]. In our study, even though all participants included in the final analysis were fully vaccinated, having only 60.2% reporting willingness to receive the booster dose raises worries about vaccine hesitancy in these groups that must be addressed by public health and effective science communication intervention.

Before the COVID-19 pandemic, people’s unwillingness to receive newly created vaccines was already a major concern [30]. Several factors, including age, gender, educational level, and risk perceptions, have been shown to drive people toward, or discourage them from, receiving vaccines [30,31,32,33].

In many countries, younger generations are much more vaccine-hesitant, which impedes immunization progress [34,35,36,37,38]. However, we overlooked the significance between age groups in our study. Hence, it is crucial to study the impact of external factors on the vaccination decision. An extensive survey by Lazarus et al. (*n* = 13,426) showed that age variations were significant in Brazil, Ecuador, Mexico, and South Africa, where older people were more inclined to follow their employer’s vaccination recommendations. However, younger people were much more likely to do so in France and the USA. Results were similar when compared to respondents from Italy, Poland, and Russia, in which younger people were more likely to express their intention to accept an employer’s vaccine recommendation [39]. Our study examined the influence of age on fear and found that more younger people had an average fear score above 5 compared to older people (23.1% vs. 13.5%, *p* = 0.003, respectively). This can be explained as the elderly are considered a high-risk population for serious COVID-19 infection, suggesting that a booster dose might be advantageous in providing them with further infection protection [40].

Furthermore, vaccination reluctance has been higher among younger women in various nations [41,42,43]. In our study, similar findings were observed, since 27.6% of females were unwilling to receive the booster dose compared with 19.5% of males. This observation was also associated with fear, since females expressed higher fear levels toward the booster dose than males (27.2 vs. 14.5, *p* <0.000001, respectively).

Moreover, Thomas and Darling reported that the educational level more strongly influences people’s willingness to obtain the vaccine. People of Asian heritage may be the sole exception, since they showed a high willingness to be vaccinated regardless of their educational level. In addition, people with lower levels of education were less trusting of the vaccines [44]. Moreover, in Ecuador, France, Germany, India, and the United States, highly educated people said they would accept a vaccine. Higher education was linked to lower vaccination acceptability in Canada, Spain, and the United Kingdom [39]. According to the three investigated educational levels, we did not notice a significant change in the willingness to accept a booster dose or the fear level. However, our observation should not be generalized due to the smaller sample size of the primary and secondary groups (*n* = 162). Several publications show that relying on social media can be dangerous, as they create an environment for spreading misleading information that may affect public mental health [45,46,47,48].

Regarding vaccine hesitancy and its association with the area of residence (urban vs. rural), the results from a survey of 1676 adults living in the United States, including Alaska and Hawaii (including interviews from 298 Hispanic adults and 390 non-Hispanic Black adults), showed that those who live in rural areas are still more likely to be vaccine-hesitant than those who live in suburban and urban areas [49]. Moreover, a study of 26,241 children from 23 countries was conducted between 2010 and 2018 in sub-Saharan Africa to determine what characteristics are linked to a child’s full immunization status in rural and urban areas. The findings stated that over half of youngsters in urban areas (52.%) were fully immunized, but 59.3% of rural children were not, due to many factors [50]. Our study did not report any significant differences between rural and urban residents regarding willingness to receive a booster dose. However, people in rural areas reported a significantly higher level of fear. More awareness is required for rural areas, and further research should be conducted to identify the reasons behind the fear in rural areas.

SARS-CoV-2 infection status has also influenced people’s decisions. After at least one vaccination dose, infected participants had an almost 50% lower probability of accepting a booster dose. At the same time, those who were regularly or irregularly vaccinated with the influenza vaccine had higher odds (OR = 1.36 and OR = 2.02, respectively) of accepting a booster dose. These findings imply that previous vaccination experience dramatically influenced people’s decisions to receive a booster dose.

The immunosuppressed and chronic patients were willing to receive the vaccine but reported a higher fear level than normal individuals. However, immunosuppressed patients with a fear level above 5 had higher odds of rejecting the vaccine, as shown in Table 5. Many countries decided to administer a booster dose to immunocompromised individuals [51]. Research shows that booster doses improve immune response and associated protection in patients, such as transplantation recipients [21,52]. Despite the emergence of highly pathogenic variants with higher potency to generate more severe infections, the reported results reaffirm the high rate of vaccination effectiveness towards severe COVID-19 infection, despite a considerable decline in immunity against SARS-CoV-2 infection [53]. This may explain the elevated fear levels, especially in high-risk groups such as immunosuppressed and chronic patients.

A tendency to favor American-based vaccines was observed in a survey released in Saudi Arabia, Lebanon, Jordan, and Iraq, since 45.2% of the participants favored BNT162b2, whereas 30% were unfamiliar with different types of COVID-19 vaccines [23]. Almost 40% of the respondents reported willingness to receive BNT162b2, followed by AZD1222 (18.34%), whereas 17.82% reported they did not know. This also explains why some Saudi citizens had to take second doses of different vaccines due to increased demand for one type, and it might also be linked to their elevated fear levels. According to Dror et al., the rapid development of vaccines increased individuals’ hesitancy to accept them [54]. Sallam et al. highlighted that lack of trust in vaccinations and producers were major factors underlying vaccine hesitation in the MENA region [55]. MENA region countries implemented several methods to encourage vaccination. The Egyptian government allowed only vaccinated individuals to enter governmental facilities [56,57]. In addition, they increased the number of vaccination centers in rural and urban areas and conducted virtual events through social media and local media to increase awareness. Community awareness campaigns were also launched in several locations with low vaccination frequency; many volunteers and healthcare professionals were involved in such campaigns and communicated with the public about the importance of COVID vaccination [58,59,60,61,62]. Still, healthcare professionals and scientists need to invest more effort toward addressing the safety of the vaccines, and more efficacy studies need to be simplified for the public.

Finally, it is worth mentioning that this study exhibits some strengths and potential limitations. First, regarding time constraints, the timing of the study was essential to achieve proper interventions since people in the MENA region had started receiving the booster dose. Second, we implemented strict inclusion and exclusion criteria to obtain accurate results. Third, the sample size covered five different MENA region countries to provide a general overview of this geographic location. Concerning the limitations, self-reporting of the fear scale might not be accurate and may reflect misreporting. Additionally, we asked participants to rate the side effects on a Likert scale; however, we did not know what these side effects were —a topic that can be further addressed in future studies. Some groups were underrepresented, such as people with different educational backgrounds (primary and secondary). Additionally, the anonymity of the survey may have attracted individuals interested in manipulating the data; even though we removed all responses below 1 min in length to prevent any usage of Macros (survey filling bots), this point still needed to be declared for transparency. Even after extending the collection time, the response rate was low, allowing only a 5% error margin, which could have been improved if we had a higher response rate. Moreover, the national coordinators distributed the survey using each country’s most commonly used social media channels; hence, the design effect should have been considered to calculate the sample size per country. Further research can benefit from mailing lists or by promoting their surveys on popular local websites. Finally, the responses represented personal opinions at a given point in time that did not fully reflect the respondents’ future decisions. As we mentioned, several factors could affect that, including the prospective status of the COVID-19 pandemic and the spread of the infodemic.

## 5. Conclusions

The main reasons to refuse the booster dose were uncertainties over their safety, belief that the booster dose is unnecessary, and side effects associated with previous COVID-19 vaccine doses. Since the introduction of booster doses has been firmly evidence-driven, efforts to improve public awareness should target the population groups in greatest need through effective scientific communication to overcome vaccine hesitancy and fear.

## Figures and Tables

**Figure 1 vaccines-10-01270-f001:**
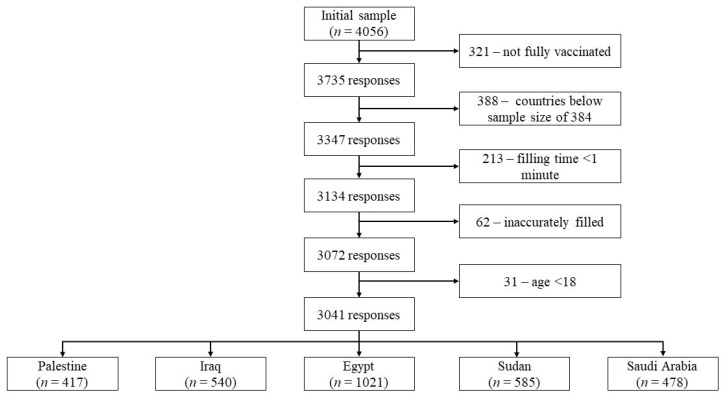
Inclusion/exclusion process for the responses.

**Figure 2 vaccines-10-01270-f002:**
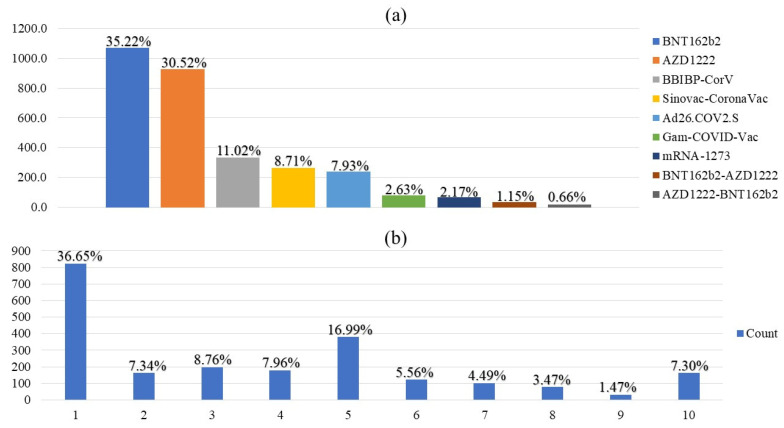
(**a**) Frequency and percentage of COVID-19 vaccination (*n* = 3041); (**b**) Fear count and percentage on a 10-point Likert scale (*n* = 2248; 10 is the highest fear score).

**Figure 3 vaccines-10-01270-f003:**
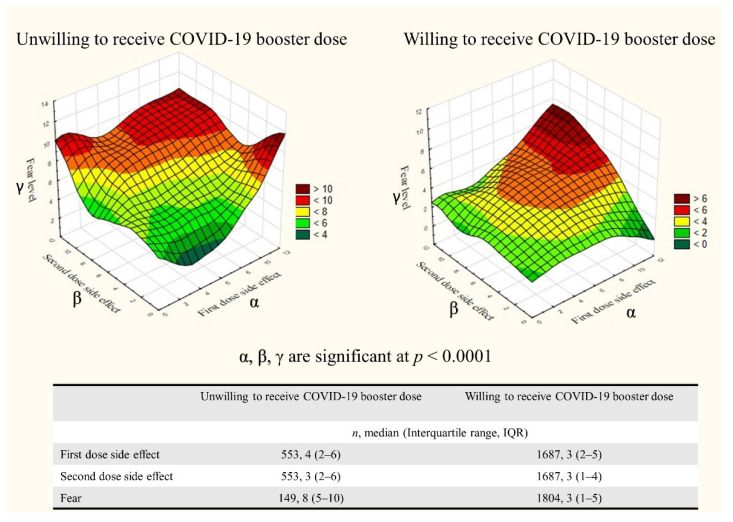
Differences between the side effects of the first and second dose of COVID-19 vaccine (α and β, respectively) and fear (γ) between individuals (willing vs. unwilling) to receive the booster dose.

**Figure 4 vaccines-10-01270-f004:**
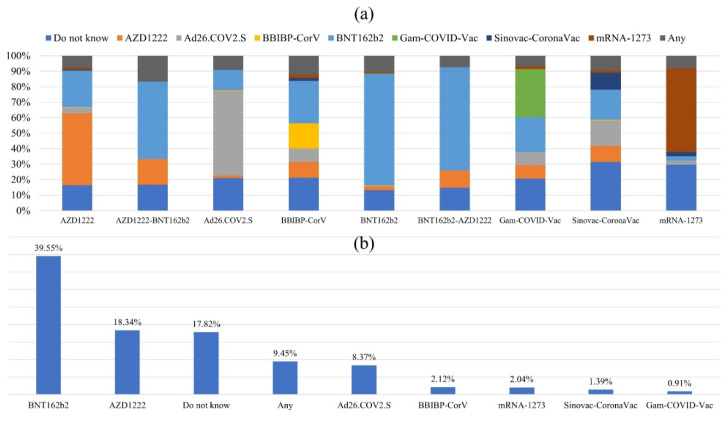
(**a**) The first choice of the specific COVID-19 vaccine booster dose among respondents earlier vaccinated with AZD1222, AZD1222-BNT162b, Ad26.COV2.S, BBIBP-CorV, BNT162b, BNT162b- AZD1222, Gam-COVID-Vac, Sinovac-CoronaVac, mRNA-1273, AZD1222, and Ad26.COV2.S (*n* = 2302); (**b**) frequencies of top-selected booster doses regardless of the previously vaccinated dose (*n* = 2306).

**Table 1 vaccines-10-01270-t001:** Demographic characteristics of the respondents (*n* = 3041), data reported as % (*n*).

Parameter	Statistics
**Age** (years), median, interquartile range (min–max)	25.0, 21.0–36.0 (18.0–73.0)
Aged <50 years	91.5 (2784)
Aged ≥50 years	8.5 (257)
**Gender**	
Female	63.2 (1922)
Male	36.8 (1119)
**BMI** * (kg/m^2^), median, interquartile range (min–max)	24.6, 21.9–27.8, (13.6–65.7)
Underweight (<18.5)	4.7 (140)
Normal weight (18.5–24.9)	44.2 (1314)
Overweight (25.0–29.9)	34.6 (1030)
Obesity (≥30.0)	16.5 (490)
**Place of residence**	
Urban	75.4 (2294)
Rural	24.6 (747)
**Education**	
Primary	0.5 (15)
Secondary	5.2 (157)
Graduate	79.3 (2413)
Postgraduate	15.0 (456)
**Immunosuppression**	9.6 (292)
**Chronic disease**	11.9 (361)
Asthma	5.1 (154)
Cardiovascular disease	1.9 (58)
Diabetes	3.8 (115)
Chronic kidney disease	0.7 (20)
Chronic pulmonary disease	0.4 (13)
Cancer	0.03 (1)
**SARS-CoV-2 infection status**	
Infected prior to vaccination	28.8 (877)
Infected between 1st and 2nd dose	4.1 (125)
Infected after full vaccination	4.6 (139)
No history of infection	62.5 (1900)
**Influenza vaccine status**	
Vaccinated annually	5.2 (157)
Vaccinated irregularly	21.6 (658)
Never vaccinated	73.2 (2226)
**Country**	
Egypt	33.6 (1021)
Iraq	17.8 (540)
Palestine	13.7 (417)
Saudi Arabia	15.7 (478)
Sudan	19.2 (585)

* Some data are missing.

**Table 2 vaccines-10-01270-t002:** The willingness and unwillingness to receive booster COVID-19 vaccine dose in several demographic groups (*n* = 2422, excluding uncertain responses, *n* = 619).

Parameter		Willing to Receive (*n* = 1831)	Unwilling to Receive (*n* = 591)	*p*-Value *
		%		
**Age**	<50	75.5	24.5	0.88
	≥50	76.3	23.7	
**Gender**	Female	72.4	27.6	<0.0001
	Male	80.5	19.5	
**BMI**	Underweight	75.0	25.0	0.02
	Normal BMI	73.7	26.3	
	Overweight	75.5	24.5	
	Obesity	81.5	18.5	
**Place of residence**	Urban	76.1	23.9	0.33
	Rural	74.0	26.0	
**Education**	Primary and secondary	79.2	20.8	0.26
	Graduate	74.9	25.1	
	Postgraduate	77.9	22.1	
**Immunosuppression**	Yes	72.7	27.3	0.35
	No	75.9	24.1	
**Chronic disease**	Yes	77.0	23.0	0.59
	No	75.4	24.6	
**SARS-CoV-2** **infection status**	Not infected	77.8	22.2	<0.0001
	Infected prior to vaccination	75.4	24.6	
	Infected after at least one dose	61.6	38.4	
**Influenza** **vaccine status**	Vaccinated annually	85.1	14.9	0.001
	Vaccinated irregularly	79.3	20.7	
	Never vaccinated	73.7	26.3	
**Country**	Egypt	76.9	23.1	<0.0001
	Iraq	70.3	29.7	
	Palestine	68.9	31.1	
	Saudi Arabia	66.8	33.2	
	Sudan	87.2	12.8	

* Chi-squared test with Bonferroni correction.

**Table 3 vaccines-10-01270-t003:** The occurrence of fear levels in several demographic groups (*n* = 2248).

Parameter		Fear ≤5 (*n* = 1747)	Fear >5 (*n* = 501)	*p*-Value *
		%		
**Age**	<50	76.9	23.1	0.003
	≥50	86.5	13.5	
**Gender**	Female	72.8	27.2	<0.0001
	Male	85.5	14.5	
**BMI**	Underweight	84.3	15.7	0.16
	Normal BMI	76.1	23.9	
	Overweight	79.3	20.7	
	Obesity	77.2	22.8	
**Place of residence**	Urban	79.4	20.6	0.0006
	Rural	72.5	27.5	
**Education**	Primary and secondary	74.5	25.5	0.46
	Graduate	77.6	22.4	
	Postgraduate	79.6	20.4	
**Immunosuppression**	Yes	25.2	74.8	<0.0001
	No	94.8	5.2	
**Chronic disease**	Yes	81.9	18.1	0.07
	No	77.1	22.9	
**SARS-CoV-2** **infection status**	Not infected	77.2	22.8	0.26
	Infected prior to vaccination	79.6	20.4	
	Infected after at least one dose	74.2	25.8	
**Influenza** **vaccine status**	Vaccinated annually	74.8	25.2	0.047 **
	Vaccinated irregularly	74.2	25.8	
	Never vaccinated	79.1	20.9	
**Country**	Egypt	71.0	29.0	<0.0001
	Iraq	86.7	13.3	
	Palestine	82.5	17.5	
	Saudi Arabia	69.8	30.2	
	Sudan	84.4	15.6	

* Chi-squared test with Bonferroni correction. ** Bonferroni correction shows no significant difference between groups.

**Table 4 vaccines-10-01270-t004:** The univariate logistic of the significant predictors of accepting COVID-19 vaccine booster dose.

Predictor	Intercept	Standard Error	Wald Chi-Square	Sig	Exp (B)	95% CI for EXP(B)	
						Lower	Upper
**Type of vaccine**							
**Ad26.COV2.S**	0.46	0.20	5.55	0.0185	1.59	1.08	2.33
**AZD1222**	0.65	0.12	29.06	<0.0001	1.91	1.51	2.42
**BBIBP-CorV**	0.39	0.17	5.34	0.0208	1.47	1.06	2.04
**mRNA-1273**	−0.95	0.30	10.23	0.0014	0.37	0.22	0.69
**Side effects and fear**							
**First dose**	−0.11	0.02	28.86	<0.0001	0.90	0.86	0.93
**Second dose**	−0.18	0.02	74.25	<0.0001	0.84	0.81	0.87
**Ad26.COV2.S**	−0.20	0.07	8.60	0.0034	0.82	0.71	0.94
**Fear**	−0.46	0.03	190.40	<0.0001	0.63	0.60	0.67
**SARS-CoV-2 infection status**							
**Infected after at least 1 dose**	−0.78	0.15	26.04	<0.0001	0.46	0.34	0.62
**Influenza vaccination**							
**Irregular**	0.31	0.12	6.71	0.0096	1.36	1.08	1.72
**Regular**	0.71	0.25	8.17	0.0044	2.03	1.25	3.30
**Sex**							
**Males**	0.45	0.10	20.20	<0.0001	1.57	1.29	1.91
**Country**							
**Sudan**	1.02	0.18	33.53	<0.0001	2.77	1.96	3.91
**Egypt**	0.31	0.14	4.65	0.0310	1.36	1.03	1.79

**Table 5 vaccines-10-01270-t005:** The multifactorial model (backward stepwise regression) analyzes the significant predictors of accepting COVID-19 vaccine booster doses.

Predictor	Intercept	Standard Error	Wald Chi-Square	Sig	Exp (B)	95% CI for Exp (B)	
						Lower	Upper
Fear	2.25	0.21	118.74	<0.0001	9.50	6.34	14.25
Immunosuppressed	−0.89	0.26	11.73	0.0006	0.41	0.25	0.68

## Data Availability

The data presented in this study are available on request from the corresponding author.

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
