# Peer review of "Attitudes toward Receiving COVID-19 Booster Dose in the Middle East and North Africa (MENA) Region: A Cross-Sectional Study of 3041 Fully Vaccinated Participants"

_vaccines, 2022, doi:10.3390/vaccines10081270_

Round 1

Reviewer 1 Report

Major

needs careful review of the statistical tests used per occasion. The univariate analysis does not account for all these statistically significant differences in the demographic factors

Minor

"Considering population size of 100,000,000, 95% confidence intervals, and 5% 130 confidence limit. The estimated minimum sample size was 385 per country": needs careful proofing. This sentence does not make sense

Figure 1: the exclusion criterion about the minimum sample size per country is not clear

Figure 2b: would present data differently; not clear and misleading

Author Response

Reviewer 1:

needs a careful review of the statistical tests used per occasion. The univariate analysis does not account for all these statistically significant differences in the demographic factors.

Thank you, we agree that univariate alone does not account for the statistical significance; hence, we have performed multiple stepwise regression analyses and added a new table (table 5).

Table 5. The multifactorial model (backward stepwise regression) analyzes the significant predictors of accepting COVID-19 vaccine booster doses.

Minor

"Considering population size of 100,000,000, 95% confidence intervals, and 5% 130 confidence limit. The estimated minimum sample size was 385 per country": needs careful proofing. This sentence does not make sense.

Thank you, we have revised the sentence to:

OpenEpi was used to calculate the sample size according to the proportion. The following parameters were used; the population size is 100,000,000, and 5% for the confidence limit. Therefore, for a 95% confidence level, the estimated minimum sample size was 384 per country

Figure 1: the exclusion criterion about the minimum sample size per country is not clear.

Thank you for noticing this; we modified figure 1 for better clarity.

Figure 2b: would present data differently; not clear and misleading

We modified figure 2b as requested.

Reviewer 2 Report

The manuscript presents interesting information about the region which was not investigated for the purpose of COVID-19 vaccine booster dose acceptance. In my opinion, the manuscript will benefit after considering the following remarks:

Materials and Methods:

*P3-L108 – it is hard to say ‘validated survey’. Tools used in a survey may be validated. It should be clarified. The paper by Rzymski does not provide any information about validity of the tools. Validity parameters should be provided. As it was an on-line survey it is especially important to know how reliability and accuracy had been assessed. Other validity criteria seem to be interesting too.

*For fear, what was the lowest level? 1 or 10?

*More information about dates, periods of recruitment and data collection should be provided in the main test. List the social media used and how the mailing lists had been obtained – this is key information taking into account the sampling bias highly possible in this study.

*Regarding sample size: what was the feature used to assess the sample size (the frequency, OR?). It is not clear to me whether authors decided on 5% precision level for point estimates? The information provided in the manuscript does not provide any idea and does not enable to do it by other researchers.

*Statistical analysis: provide information, please, how the missing data were treated?

*There is some probability of confounding effect in this study. Add please some additional analyses to address this issue.

*Authors should list what side effects were considered in their study

Results:

*It would be useful to know what side effects were associated with unwillingness to get a booster dose of COVID-19 vaccine.

*P6-L182: What data is not shown? What is the data provided in the brackets?

*P7-Fig.3 Provide, please, clear description what is alpha, beta and gamma.

*P8-table3: p-value of 0.047 for influenza vaccine status … under Methods p-value <0.05 considered statistically significant, but here 0.047 as no significant difference – could You explain?

*P9-L219: Predictors of accepting COVID-19 vaccine booster dose. Authors provide results of univariate logistic regression – these results, however, are not free from confounding effects. Run multivariate logistic regression to minimize confounding. Other strategies are also welcomed.

*P11-L330: authors says “we implemented strict inclusion and exclusion criteria to have refined results. This sounds like manipulation … Does it mean they limited their sample in a way to get results they wanted to get??? All the data should be used and analyzed. Any strategy of data limitation for the purpose to get better, refined results are not accepted any way.

*P11-L332: fear is a subjective issue, so it is not clear to me how self-reporting of fear might not be accurate?

*Other sources of bias, especially sampling bias should be discussed here. What was the response rate? Did You implement any strategies to get better response rate?

*P11-L336: the “usage of Marcos” – this is not understandable to me … and probably to other readers, explain, please.

Conclusions

“All participants in this cross-sectional study were fully vaccinated” – this is not a conclusion, this is a consequence of the inclusion criteria. The second statement is very general and does not provide sound information.

Reviewer

Author Response

Reviewer 2:

The manuscript presents interesting information about the region which was not investigated for the purpose of COVID-19 vaccine booster dose acceptance. In my opinion, the manuscript will benefit after considering the following remarks:

We greatly thank you for the feedback; we have carefully revised our manuscript as per your suggestions.

Materials and Methods:

*P3-L108 – it is hard to say ‘validated survey’. Tools used in a survey may be validated. It should be clarified. The paper by Rzymski does not provide any information about validity of the tools. Validity parameters should be provided. As it was an on-line survey it is especially important to know how reliability and accuracy had been assessed. Other validity criteria seem to be interesting too.

Please note that Rzymski mentioned that “The scales’ internal consistency reliability was determined with Cronbach’s alpha and demonstrated good reliability of α = 0.82–0.93”.

Therefore, we have also clarified this in our manuscript as you suggested.

*For fear, what was the lowest level? 1 or 10?

Thank you for noting this. We have added to our manuscript

1 is the lowest and 10 is the highest. We also highlighted this in the figure 2 legend and inside the text.

*More information about dates, periods of recruitment and data collection should be provided in the main test. List the social media used and how the mailing lists had been obtained – this is key information taking into account the sampling bias highly possible in this study.

please note that we added the following information to the manuscript:

The survey distribution started on 6th December 2021 and was concluded on 9th January 2022. The survey was distributed to various groups, including educational, commercial, and community, on social media channels, mainly WhatsApp and Facebook.

*Regarding sample size: what was the feature used to assess the sample size (the frequency, OR?). It is not clear to me whether authors decided on 5% precision level for point estimates? The information provided in the manuscript does not provide any idea and does not enable to do it by other researchers.

Please note that we used the following link to calculate the sample size:

http://www.wwww.openepi.com/SampleSize/SSPropor.htm

we also have modified the text as follows:

OpenEpi was used to calculate the sample size according to the proportion. The following parameters were used; the population size of 100,000,000 and 5% confidence limit. Therefore, for 95% confidence level, the estimated minimum sample size was 384 per country

Please note that this method was implemented earlier by other scholars:

https://globalizationandhealth.biomedcentral.com/articles/10.1186/s12992-021-00768-3

*Statistical analysis: provide information, please, how the missing data were treated?

We added Missing data were treated by listwise and pairwise deletions.

Please note as an example:

if we asked for weight and the person wrote (I do not know my weight), then we perform pairwise. But if the person responded something like (9999999999) we perform listwise deletion because we are afraid that he/she is inaccurate filling the survey without reading carefully.

*There is some probability of confounding effect in this study. Add please some additional analyses to address this issue.

This is a great idea; we have performed multiple stepwise regression analyses, and we have added a new table (table 5).

Table 5. Multifactorial model (backward stepwise regression) analysis of the significant predictors of accepting COVID-19 vaccine booster dose.

Results:

*Authors should list what side effects were considered in their study

*It would be useful to know what side effects were associated with unwillingness to get a booster dose of COVID-19 vaccine.

We agree with you, and the original survey did not include such data. Hence, we did not ask for the side effects. It could be a good topic for further research.

*P6-L182: What data is not shown? What is the data provided in the brackets?

We are sorry for the confusion, and we meant we did not provide a figure. Hence, we removed this sentence for better clarity.

*P7-Fig.3 Provide, please, clear description what is alpha, beta and gamma.

We have modified the fig.3 legends for better clarity.

Figure 3. Differences between the side effects of the first and second dose of COVID-19 vaccine (α and β, respectively) and fear (γ) between individuals (willing vs. unwilling) to receive the booster dose.

*P8-table3: p-value of 0.047 for influenza vaccine status … under Methods p-value <0.05 considered statistically significant, but here 0.047 as no significant difference – could You explain?

I am very happy you asked this question; please note that 0.047 was the value for the Chi-squared test without Bonferroni correction; when we did Bonferroni correction, it showed there were no differences between the groups after adjustment. This highlights the importance of conducting such an analysis for better accuracy. We modified the table legend to:

** Bonferroni correction shows no significant difference between groups

*P9-L219: Predictors of accepting COVID-19 vaccine booster dose. Authors provide results of univariate logistic regression – these results, however, are not free from confounding effects. Run multivariate logistic regression to minimize confounding. Other strategies are also welcomed.

This is a great idea; we have performed multiple stepwise regression analyses and added a new table (table 5).

Table 5. Multifactorial model (backward stepwise regression) analyzes the significant predictors of accepting COVID-19 vaccine booster dose.

*P11-L330: authors says “we implemented strict inclusion and exclusion criteria to have refined results. This sounds like manipulation … Does it mean they limited their sample in a way to get results they wanted to get??? All the data should be used and analyzed. Any strategy of data limitation for the purpose to get better, refined results are not accepted any way.

Please note that we meant we analyzed all the data after filtration and removed inaccurate responses. All the deleted responses were included in Figure 1 and why we deleted them.

 For example, some people said they took only the first vaccine shot, and they reported the first side effects, which is an inaccurate response. Other people filled multiple surveys in 30 seconds using some macros, filling bots, or filling randomly. Therefore, we have removed all these inaccurate responses. Once we filtrated these responses, we directly performed the statistical analysis, and we reported the results as it is. So, we did not manipulate our results but cleaned our data to prevent any manipulation. To help the reader understand better, we changed the word refined to accurate.

That is why we also added earlier this sentence:

Besides, the anonymity of the survey may attract individuals interested in manipulating the data; even though we removed all responses below 1 minute to prevent any usage of Macros (survey filling bots), this point still needed to be declared for transparency

*P11-L332: fear is a subjective issue, so it is not clear to me how self-reporting of fear might not be accurate?

We totally agree with you. We have added it to the limitation because the fear might vary from one person to another, especially if they have to use a scale (5 for some could be 4 for another). Also, the fear status might change from one time to another due to many environmental interactions. For example, when a person sees his family or friends getting vaccinated and feeling better after, this fear might change later.

*Other sources of bias, especially sampling bias should be discussed here. What was the response rate? Did You implement any strategies to get better response rate?

Please note that we added the following paragraph:

Even with extending the collecting time, the response rate was low, allowing only a 5% error margin, which could have been improved if we had a high response rate. Further research can benefit from mailing lists or promoting their surveys on famous local websites.

*P11-L336: the “usage of Marcos” – this is not understandable to me … and probably to other readers, explain, please.

We apologize for the mistyping, and it is Macros (survey filling bots).

Conclusions

“All participants in this cross-sectional study were fully vaccinated” – this is not a conclusion, this is a consequence of the inclusion criteria. The second statement is very general and does not provide sound information.

Thank you, we have removed these sentences.

Reviewer 3 Report

This is an interesting paper on the willingness to be vaccinated for the COVID-19  booster. The selection of samples from five countries is a strength.  The analysis has effectively identified important attributes influencing the vaccination behavior as well as the attitudes toward the preventive action.

The paper could elaborate on results presented on the country's variations in preventive behavior such as booster taking. Otherwise, I think that this is a well-organized study.  

Author Response

Thank you for your pleasant feedback. We added this to the text:

MENA region countries implemented several methods to encourage vaccination. Egyptian government allowed only vaccinated individuals to enter governmental institutes [55,56]. In addition, they increased the number of vaccination centers in rural and urban areas and conducted virtual events through social media and local media to increase awareness. Community awareness campaigns were also launched in several locations with low vaccination frequency; many volunteers and healthcare professionals were involved in such campaigns and communicated with the public about the importance of covid vaccination [57–61].

Round 2

Reviewer 2 Report

Dear Editor, Dear Authors,

The manuscript has been improved after implementation of some of my comments. In my opinion before the publication some minor issues should be also addressed. These are as follows:

1) The authors omitted accuracy measures in their comments about tool validity - address this issue too.

2) Regarding sample size ... the key information here is the hypothesized frequency of the outcome in the population, which was not provided by the authors. Moreover, the sampling was not random in the presented study - was limited by the use of social media channels - thererfore a design effect should be considered.

3) side effects: add, please, a comment that this type data was not collected to the content of the manuscript.

Best,

Reviewer

Author Response

1) The authors omitted accuracy measures in their comments about tool validity - address this issue too.

I am sorry if I omitted this point; I might have mixed the terms the accuracy and reliability. Please note that we highlighted they measured the Cronbach alfa for the survey reliability and added it to the manuscript. Regarding the accuracy, I emailed the corresponding author of the original survey, Dr. Dr.Rzymski if they had done any accuracy for their tool, and he told me that accuracy refers to "sample representativeness" and the term "survey accuracy" is "the extent to which a questionnaire result represents the attribute being measured in the group of interest or population". Then he told me that it is about the sample size and asked to use this website:

http://www.raosoft.com/samplesize.html

Hence, I am not sure if they conducted any test for accuracy.

Therefore, regarding the accuracy of the sample representative, we had 3041 fully vaccinated persons from MENA region, which can provide us with a broader image of their desire to accept the vaccine, especially in MENA region countries that share the same traditions, thoughts, and concepts.

I also added this sentence to the manuscript limitation:

Finally, the national coordinators distributed the survey using each country's most commonly used social media channels; hence, the design effect should have been considered to calculate the sample size per country to improve the accuracy of the results.

I hope I have clarified this point.

3) side effects: add, please, a comment that this type data was not collected to the content of the manuscript.

Please note that I added these limitations:

We asked the participants to rate the side effects on the Likert scale; however, we did not know what these side effects – a topic that can be further addressed in future studies

2) Regarding sample size ... the key information here is the hypothesized frequency of the outcome in the population, which was not provided by the authors. Moreover, the sampling was not random in the presented study - was limited by the use of social media channels - therefore, a design effect should be considered.

Kindly note that hypothesized frequency was 50% since we did not know the exact percentage. According to this website:

https://ourworldindata.org/covid-vaccinations

Egypt is 35%, KSA is 75%, Sudan is unknown, Iraq 18%, and Palestine 35%. Hence, we collected more sample required by these values.

Regarding the sampling, originally, the sample size calculation was performed using the design effect of 1 since we did not intend to send the survey via social media channels. But as I kindly reported earlier, we had national co-coordinators responsible to speared the survey in their region so they used the most common way in their country, for example, in KSA they used WhatsApp and Twitter, in Palestine, they used Facebook groups, in Egypt they used Facebook and telegram. Usually, mailing list is not common in MENA region. Still, I agree with you that design effect could be applied for the entire MENA region and we still have good sample size but we can not apply it per country because some countries will have a low number of participants. So, we addressed this in the limitations as well.

We added the following to our manuscript:

OpenEpi was used to calculate the sample size according to the proportion. The following parameters were used; the population size is 100,000,000, and 5% for the confidence limit. The hypothesized frequency was set to 50%, and the design effect is 1. Therefore, for a 95% confidence level, the estimated minimum sample size was 384 per country

In the limitation:

Finally, the national coordinators distributed the survey using each country's most commonly used social media channels; hence, the design effect should have been considered to calculate the sample size per country to improve the accuracy of the results.